# A Two-Step Approach to Overcoming Data Imbalance in the Development of an Electrocardiography Data Quality Assessment Algorithm: A Real-World Data Challenge

**DOI:** 10.3390/biomimetics8010119

**Published:** 2023-03-13

**Authors:** Hyun Joo Kim, S. Jayakumar Venkat, Hyoung Woo Chang, Yang Hyun Cho, Jee Yang Lee, Kyunghee Koo

**Affiliations:** 1Department of Anesthesiology and Pain Medicine, Anesthesia and Pain Research Institute, Severance Hospital, Yonsei University College of Medicine, Seoul 03722, Republic of Korea; 2Department of Thoracic and Cardiovascular Surgery, Seoul National University Bundang Hospital, Seoul National University College of Medicine, Gyeonggi-do, Seongnam-si 13620, Republic of Korea; 3Department of Thoracic and Cardiovascular Surgery, Samsung Medical Center, Sungkyunkwan University College of Medicine, Seoul 06351, Republic of Korea

**Keywords:** electrocardiography, signal quality, machine learning, random forest

## Abstract

Continuously acquired biosignals from patient monitors contain significant amounts of unusable data. During the development of a decision support system based on continuously acquired biosignals, we developed machine and deep learning algorithms to automatically classify the quality of ECG data. A total of 31,127 twenty-s ECG segments of 250 Hz were used as the training/validation dataset. Data quality was categorized into three classes: acceptable, unacceptable, and uncertain. In the training/validation dataset, 29,606 segments (95%) were in the acceptable class. Two one-step, three-class approaches and two two-step binary sequential approaches were developed using random forest (RF) and two-dimensional convolutional neural network (2D CNN) classifiers. Four approaches were tested on 9779 test samples from another hospital. On the test dataset, the two-step 2D CNN approach showed the best overall accuracy (0.85), and the one-step, three-class 2D CNN approach showed the worst overall accuracy (0.54). The most important parameter, precision in the acceptable class, was greater than 0.9 for all approaches, but recall in the acceptable class was better for the two-step approaches: one-step (0.77) vs. two-step RF (0.89) and one-step (0.51) vs. two-step 2D CNN (0.94) (*p* < 0.001 for both comparisons). For the ECG quality classification, where substantial data imbalance exists, the 2-step approaches showed more robust performance than the one-step approach. This algorithm can be used as a preprocessing step in artificial intelligence research using continuously acquired biosignals.

## 1. Introduction

Despite the emergence of artificial intelligence (AI) technology in medicine, biosignal-based AI is relatively unexplored compared to other fields. There are several reasons as to why it has been difficult to find breakthrough achievements thus far. The first reason is probably because the size of the data is relatively large compared to data sizes in other medical areas. The amount of computation required to develop an algorithm based on biosignals is large, and a complex algorithm is needed to handle these time-series data. However, another significant obstacle in biosignal research is that a tool is needed to distinguish acceptable quality data from unacceptable quality data. Despite the large amount of biosignal data, it is necessary to determine the normally recorded section in a dataset before proceeding to algorithm development. Even when evaluating the performance of a biosignal-based algorithm, the quality of the test data should be guaranteed first to validate the algorithm.

Among various biosignals, electrocardiography (ECG) is the most important, as it shows cardiac activity by measuring minute electric potential. ECG signals contain much information about heart conditions that are still unknown. However, ECG is frequently contaminated by artifacts such as electrical interference, baseline wander, electromyogram, or poor electrode contact [1,2]. Occasionally, ECG signals contain pacing spikes caused by pacemakers that give rise to extremely large amplitudes. Despite state-of-the-art internal filter algorithms in commercial patient monitors, there are poor-quality ECG recordings from patient monitors due to external factors beyond our control. Considering the large amount of data required for developing an AI algorithm, it is impossible to manually select high-quality data. Automatic ECG quality assessment is an essential prerequisite not only for algorithm development but also for using the algorithm in clinical practice.

There have been attempts to automate ECG quality assessment. Traditionally, studies have focused on feature extraction or hand engineering of fiducial and non-fiducial features [3,4,5,6,7,8]. ECG denoising-based strategies have focused on suppressing noise and artifacts in ECG recordings. Generally, denoising methods are capable of suppressing noise and artifacts from raw ECG signals. However, there are concerns about the distortion of innate ECG characteristics such as amplitude, duration, and interval [9,10]. A signal quality index (SQI)-based strategy was introduced before the popularization of feature extraction, and in this approach, the recorded ECG segments were categorized as having acceptable or unacceptable quality [11,12].

However, due to the availability of massive biosignal datasets [13,14,15,16], researchers have recently used many deep learning techniques in the analysis of various ECG datasets [17,18,19]. During the process of developing a noninvasive estimator of arterial blood pressure based on ECG and other signals, we needed a signal quality classifier for each signal. This was the most important motivation for this study. In the process of developing a real-time decision support system based on biosignals, we investigated methods for ECG signal quality classification using machine learning and deep learning approaches.

## 2. Materials and Methods

The study design is shown in Figure 1. The Institutional Review Board of Seoul National University Bundang Hospital (SNUBH) (IRB No. B-2109-706-307) and Severance Hospital (SEVH) (IRB No. 4-2022-0450) approved this study, and individual patient consent was waived, as prospectively collected vital sign waveform signal databases from both hospitals were used. This study was conducted in accordance with the Declaration of Helsinki and the Harmonized Tripartite Guideline for Good Clinical Practice from the International Conference on Harmonization. All methods were performed in compliance with the relevant guidelines and regulations.

The deep learning algorithm was developed and tested with Python 3.5.6 (Anaconda Inc., Austin, TX, USA), including the Keras 2.2.2 and TensorFlow 1.10.0 libraries. Four approaches were developed and tested: (1) a one-step, 3-class random forest (RF) approach (1-step RF), (2) a one-step, 3-class 2-dimensional convolutional neural network (2D CNN) (1-step 2D CNN), (3) a two-step binary sequential RF approach (2-step RF), and (4) a two-step binary sequential 2D CNN approach (2-step 2D CNN).

### 2.1. Data Collection

Training and validation data were collected from 18 patients in the surgical intensive care unit of SNUBH from January to February 2022. External test data were collected from 5 operating rooms in SEVH from December 2021 to January 2022. ECG signals from 18 patient monitors (IntelliVue MX700, Philips, Amsterdam, The Netherlands) in SNUBH and 5 patient monitors (IntelliVue MX700, Philips, Amsterdam, The Netherlands) in the operation rooms of SEVH were collected using the Vital Recorder wireless transmission and collection system [20]. Either 3 or 5 electrodes were used, and the ECG signals from leads II, III, or aVF reconstructed using the patient monitor were obtained at a 250 Hz frequency and were segmented into lengths of 20 s. Therefore, a segment was a vector of 5000 values (250 Hz × 20 s).

### 2.2. Annotation by Expert Physicians

A simple annotation tool with a graphical user interface was developed. The annotation tool was run in Jupyter Notebook. It displayed a 20-s ECG segment with the *Y*-axis ranging from −5 mV to 5 mV. Two expert physicians (H.W. Chang and H. J. Kim) inspected the shape of the ECG segment and determined which class the segment should be categorized into by clicking the buttons shown on the tool. To reduce inter-annotator bias, the two experts participated in vigorous discussions to synchronize the annotations. The segments where the experts assigned different labels (less than 2% of all segments) were separately evaluated in a research meeting, and the final label was determined.

### 2.3. Data Cleaning and Preprocessing

The data cleaning and preprocessing steps were performed using a separate preprocess code (dataingestor.py, Figure 2, Appendix A). Twenty-s ECG segments that were fully blank or partially blank for >0.5 s were excluded and did not enter the training or test process. Extremely high or low ECG segment values were set to 0 using ±5 mV cutoff values, as abruptly appearing extreme values (>100 mV or <−100 mV) were found during the inspection of the data. Additionally, those segments were then interpolated to replace the suddenly appearing single zero values. Only the ECG segments that passed these preprocessing steps were used for training or testing.

### 2.4. Definition of ECG Quality Classes

The patient monitors used in our study were equipped with one of the state-of-the-art signal filter algorithms. Therefore, the ECG signals used in this study were processed with the internal filter algorithm of the patient monitor. Nevertheless, not all the baseline sways or noise could be removed depending on the recording circumstances.

We classified the ECG segments into three quality classes: acceptable, unacceptable, and uncertain. When a segment was clean and contained almost no flaws in recording, the segment was categorized as acceptable. A segment that contained a large flaw (i.e., extreme baseline sways or noise that rendered the QRS complexes indistinguishable) for any duration was categorized as unacceptable. The segments that were almost acceptable in recording quality but had mild degree noise or subtle baseline sways were categorized as uncertain. In uncertain segments, at least the location of the QRS complexes had to be discernible in all areas. The presence of noise was a factor in quality classification, but even a data segment without noise could be classified as unacceptable when the quality of the recording was poor. Typical examples of each class are shown in Appendix A.

### 2.5. Data Sampling for Training

As described above, only the SNUBH SICU data were used for training and validation. Data selection for training was completely random. Due to the availability of a large number of acceptable samples, which led to data imbalance among the 3 classes, training was performed only on an equal number of training data from each class selected via random sampling (Figure 3). The training–validation split was set to 0.2.

### 2.6. One-Step 3-Class Approaches

The logic of the one-step, 3-class classifier is shown in Figure 3a. For the one-step, 3-class approaches, an RF and a 2D CNN classifier were developed. The RF classifier was operated on the features extracted with a separate feature extraction algorithm (which is explained in Section 2.8), and the 2D CNN classifier used the data transformed with a short-time Fourier transformation (STFT).

### 2.7. Two-Step Binary Sequential Approaches

We also tested 2-step approaches to address data imbalance. We utilized two binary classifiers sequentially for the 2-step approaches (Figure 3b). The first classifier was trained exclusively on unacceptable and acceptable segments (not on uncertain segments), and the second classifier was trained exclusively on unacceptable and uncertain segments.

During the inference step, a candidate ECG segment was classified into acceptable vs. other (unacceptable or uncertain) by the first classifier. When the segment was classified as other, it was then classified as unacceptable or uncertain by the second classifier. As in the one-step 3-class approaches, RF and 2D CNN classifiers were used in the two-step binary sequential approaches. The RF classifiers used the extracted features, and the 2D CNN classifiers used fast Fourier transform (FFT)-applied data as input.

### 2.8. Feature Extraction

For the one-step and two-step RF approaches, we extracted time and frequency domain features. These features were developed and defined based on careful observation of typical segments from each class. Additionally, concepts of mathematical generalization approaches that have been used for nonbiological waveform signals were partially adopted. The fundamental details and pseudocodes for feature extraction are shown in Appendix A. A bandpass filter with cutoff frequency [0.5, 8.0] was applied to the ECG signals to eliminate very low frequencies prior to peak detection. ECG peaks were detected using the heartpy python package. FFT and STFT were applied using the scipy fft and scipy.signal.stft functions to extract frequency domain features. Altogether, 83 features were extracted from each ECG signal. Since the ECG segments were 5000 samples long, features were extracted using this approach and further aggregated using the mean and standard deviations (rather than from the entire segment alone). Features were also extracted by negating the ECG signal for two reasons: (1) Peak detection routines return only peaks and not troughs. (2) There were sudden changes in the polarity of the ECG signals.

### 2.9. Training of the Classifiers

In both the one-step and two-step approaches, we used the RF classifier in the sklearn python package. The RF classifiers were trained on the features selected with scipy’s SelectKBest method, where k was set to 43. In the RF classifiers, the n_estimators parameter of the RF model was incremented in each training step by 50 to update the model with new samples selected from the acceptable class. The bootstrap and out-of-bag sample parameters were set to True, and the warm_start parameter was set to True. The list of the 43 selected features is shown in Table 1. The simplified codes for training the RF classifier can be found in Appendix A.

The 2D CNN classifiers in the one- and two-step approaches were partially inspired by the approach shown in the paper by Yoon et al. [18]; this approach is a 1D CNN binary classifier. The structure of the 2D CNN classifier in the one-step 3-class approach is illustrated in Figure 4a. The structure of a binary classifier in the 2-step binary sequential 2D CNN approach is illustrated in Figure 4b. Using a sliding window size of length 10 and an overlap of 5 samples, every 20-s ECG segment was converted into a 2D matrix and reshaped into the dimension (80, 100). The architecture consisted of sequentially stacked convolutional layers with a geometrically decreasing number of feature maps starting from 100 down to 25, with correspondingly decreasing kernel sizes. Average pooling was used in descending order of the pool size, i.e., 10, 5, and 3, and the stride was set to 3. The ReLU function was used in all the convolutional layers with the alpha parameter set to 0.002. The keep probability for the dropout layers was set to 0.5. Outputs from the final convolutional layer were passed through three dense layers stacked with a decreasing number of neurons (128, 64, 32, 16) followed by the final dense layer with the softmax function to obtain the 3-class or binary classification output. L2 regularization was used in the convolutional and dense layers, and alpha was set to 0.01. Categorical and binary cross entropy were selected as the loss functions for the 1-step and 2-step approaches, respectively. The Adam optimizer was used with a default learning rate of 0.001, a batch size set to 50 and epoch number set to 100 to train the models.

The training scheme of the three classifiers (the three-class classifier in the 1-step approach, the first-step classifier in the 2-step approach, and the second-step classifier in the 2-step approach) are shown in Figure 5. During training, the same number of data were randomly sampled repeatedly (nRepetitions = 20) from each class, and 20% of the training data were set aside for cross-validation in every iteration of the training step.

### 2.10. Training Stop Condition of the 2D CNN

The early stopping criterion monitoring function val_loss was used to stop 2D CNN training with restore_best_weights set to True. Precision and recall scores on the validation data were computed in each training step. The model with the maximum precision and recall scores on the held-out training data was chosen as the final model.

### 2.11. Performance Test on the External Data

The four algorithms developed using SNUBH data were tested on the SEVH data. The data from SEVH were preprocessed in the same way. Moreover, features were extracted for RF approaches, and STFT or FFT were applied for the 2D CNN approaches. Confusion matrices for the original three-class classification model (3 × 3) were drawn for all four approaches. As the most important goal of the algorithm was to obtain the highest precision in choosing the data that are ‘usable’ for the development of the other AI-based tools, we thought that the performance should be evaluated by merging the acceptable and uncertain classes if we consider the proximity in the data quality of these two classes. Therefore, additional 2 × 2 confusion matrices were drawn; the 2 × 2 confusion matrices assumed that there were two classes (acceptable + uncertain and unacceptable). Precision, recall, and accuracy were calculated from the 3 × 3 and 2 × 2 confusion matrices.

## 3. Results

### 3.1. Preprocessing and Data Class Balancing

In the training and validation datasets, 2487 of 33,614 segments (7.4%) were excluded by preprocessing, and 275 of 10,054 segments (2.7%) were excluded in the external test dataset (Figure 1). The distributions of the classes were significantly different between the training/validation and test datasets; a distribution of 29,606/828/693 was used in the training/validation dataset, and a distribution of 8135/711/933 was used in the test dataset for the acceptable/unacceptable/uncertain classes, respectively (*p* < 0.001).

### 3.2. Training of the 2D CNN Classifiers

All three 2D CNN classifiers (the one-step, three-class classifier and the first and second binary classifiers in the two-step approach) reached the training stop condition in less than 20 training steps. The loss graphs can be found in Appendix A. The codes for 2D CNN training can be found in Appendix A.

### 3.3. Performance Metrics on the External Data

The confusion matrices obtained by applying four different approaches on the test dataset are shown in Figure 6. The raw numbers can be found in Appendix A. The 3 × 3 confusion matrices were drawn by using the original three classes (acceptable, unacceptable, and uncertain), and the 2 × 2 confusion matrices were drawn assuming there were only two classes (acceptable + uncertain and unacceptable). The precision and recall scores of each class and the overall accuracy were calculated from the 3 × 3 and 2 × 2 confusion matrices.

The performance metrics from the 3 × 3 confusion matrices are shown in Figure 7a. In the 3 × 3 confusion matrix, the overall accuracy was higher for the 2-step approaches, but the most important metric—precision in the acceptable class—was above 0.9 for all four approaches. However, the next important measure, recall in the acceptable class, was generally higher for the two-step approaches: 1-step (0.77) vs. 2-step RF (0.89) and 1-step (0.51) vs. 2-step 2D CNN (0.94) (*p* < 0.001 for both comparison). Precision in the unacceptable class was also higher for the two-step approaches, but the recall in the uncertain class was lower for the two-step approaches.

The performance metrics for the 2 × 2 confusion matrices are shown in Figure 7b. In the 2 × 2 confusion matrix, the overall accuracy was similar in the four classes, and the most important metric—precision in the acceptable + uncertain class—was above 0.95 for all four approaches. The next important measure, recall in the acceptable + uncertain class, was also above 0.9 for all four approaches. However, precision in the unacceptable class was generally higher for the two-step approaches: one-step (0.47) vs. two-step RF (0.72) and one-step (0.37) vs. two-step 2D CNN (0.71) (*p* < 0.001 for both comparisons).

## 4. Discussion

Biosignal data from wearable devices or continuous patient monitoring systems used in hospitals need to determine the ‘data usability’ before the data are used for training an AI-based algorithm or for inference by the developed algorithm. The data of interest in this study are continuously collected ECG signals. According to the expert annotation, 80–90% of the 20-s ECG segments were categorized as acceptable (almost flawless recordings) in both the training/validation and test datasets. However, approximately 10–20% of the data were almost completely unusable or had flaws to varying degrees. Such defects occur for various reasons, and a number of studies have already been conducted on artifacts that degrade the quality of ECG data. However, before listing complex engineering terms, such problems in ECG recordings are quite natural considering the acquisition environment of continuously collected ECG signals. Patients do not remain still but move actively or passively. During nursing or bedside procedures, some ECG electrodes are often detached or placed in a poor contact state.

Therefore, this has been a very important problem in the development of AI algorithms for ECG signals. There have been several studies on this subject [21]. Most of the previous studies applied beat-by-beat segmentation or defined data segments with lengths of 3–5 s or up to 10 s. When beat-by-beat segmentation is used, the expected ECG unit shape is relatively fixed, so it may be slightly easier to determine the quality of the signal. However, the tolerance of beat-by-beat segmentation to various arrhythmias that can be encountered in the hospital setting is low. In other words, beat-by-beat segmentation can determine the usability of ECG data on the premise that more than 99% of the input data are from nonarrhythmic subjects. Therefore, it might not be appropriate to perform beat-by-beat segmentation for ECG signals obtained in hospital settings. Furthermore, using ECG data with segments that are less than 10 s in length might still be an option for the evaluation of ECG signals. However, if other biosignals are used together with ECG signals, such a short signal unit may cause problems. Biosignals obtained from patient monitors have time differences, and in the case of other biosignals, it is often not easy to determine the usability from only a segment that is less than 10 s. Then, how can the usability of a longer segment be determined based on the continuously appearing ‘usable’ determination of short segments? In fact, this method cannot adequately respond to artifacts that may exist at both ends of short data segments. To ensure that high quality signal recording was maintained steadily for a sufficiently long period, we use a period of 20 s for each ECG segment.

There are several open biosignal datasets, including MIMIC III [14]. Such datasets help researchers develop algorithms, but it is somewhat questionable whether these data are in the same raw data format as those obtained directly from patient monitoring devices. Whether algorithms developed using such datasets can be generalized to modern real-world datasets around the globe has not yet clearly been established. Hence, we developed an algorithm from the raw data phase obtained from patient monitoring devices. When an ECG signal obtained from a patient monitoring device was segmented into 20-s intervals, 3–7% of the data were discarded during preprocessing due to a blank section of at least 0.5 s of the 20-s signal. These data can be regarded as unacceptable without being evaluated by the algorithm. In the preprocessing step, abruptly appearing spikes were removed, and interpolation was applied. This preprocess is as important as the algorithm itself and must be included as a part of the development of an AI algorithm that uses continuously monitored ECG data in the future.

The second difficulty was extreme class imbalance. Even though the experts made more than 30,000 annotations on the training/validation dataset, most of the data belonged to the acceptable class, and less than 5% of the data were in the unacceptable and uncertain classes. This caused difficulty in improving the performance on the test dataset. First, we developed one-step, three-class approaches, but as shown in Figure 7a, recall in the acceptable class, precision in the unacceptable class, and precision in the uncertain class were all unsatisfactory. The precision in the unacceptable and uncertain classes for the one-step, three-class approaches was severely affected by the higher number of acceptable segments that were misclassified as unacceptable or uncertain. Even for the metrics recalculated by merging the acceptable and uncertain classes, the performance of the one-step approaches was unsatisfactory, and the precision scores of all unacceptable classes were less than 0.5. Although the previous literature has attempted to address data imbalance in ECG datasets [22,23], these studies have specifically addressed arrythmia and adopted heartbeat segmentation-based approaches. For these reasons, we devised a two-step approach.

In the two-step binary sequential approaches, the first step used a classifier that distinguished between the acceptable class and the other classes; to do this, the classifier was trained with only acceptable and unacceptable class data, that is, the classes that are more clearly different from each other. Data from the uncertain class were not used for the training of this classifier. This is one of the reasons we categorized the data into three classes rather than two classes (acceptable vs. unacceptable). The number of previous studies that utilized three-class classification is relatively small [6]. There is a clearly discernible or distinguishable difference between the acceptable and unacceptable classes, but there is a clear gray zone between them. In our pilot study, data labeling was binary, acceptable and unacceptable. From the labeling physician’s viewpoint, the presence of ambiguous data that were difficult to categorize into either an acceptable or unacceptable class was a considerable issue. This adversely affected the accuracy and consistency of labeling. The uncertain class data are those that are similar to the acceptable class data but have small flaws in recording. We found that including uncertain class data disturbs the training of the first classifier in the two-step approaches in the preliminary experiment. We believe that this discrepancy is reasonable, as the purpose of this classifier was to distinguish acceptable class data from the rest, and for that purpose, there was no rule that data of all three classes must be used during training.

As mentioned earlier, the most important performance indicator of this algorithm was the precision score of the acceptable class, and the recall score of the acceptable class was the next important metric. The importance of the precision score of the acceptable class arises from the purpose of this algorithm. When the algorithm receives a new dataset as input and determines the usability, it is most important to minimize the amount of data that are not actually usable but categorized as usable (false usable). If there is a substantial amount of unusable data among the data categorized as ‘usable’, it can become a significant problem when developing a diagnostic support algorithm using these training data. The recall score of the acceptable class was the next important measure, as we wanted to maximize the utilization of the usable data.

In the performance evaluation of an algorithm, it is very important to use separate test data that do not overlap with the training data. The simplest and most reliable way to achieve separation is to use test data from another hospital. Separating the training/validation data from the test data can reduce the influence of overfitting that can arise from pattern recognition by patient overlap or hospital-specific environments. Perhaps the greatest strength of this study lies here.

After developing four three-class classification algorithms, we combined the acceptable and uncertain classes of the three-class results so that the performance of the algorithms could be evaluated in the form of binary classifiers (Figure 6 and Figure 7). We did this because the criterion for the decision of an acceptable class was too strict. If this strict criterion was applied in the determination of ‘usable’ data, all the uncertain class data with very small flaws would be discarded. This is impractical because the purpose of this study is to correctly select as much usable data as possible. Therefore, we also evaluated the precision and recall scores of the merged acceptable + uncertain class.

For the RF approaches, features were extracted based on observing hundreds of segments from all three classes. In general, the acceptable segments tended to be very regular and evenly spaced with no or very few extremely high or low values, while unacceptable segments tended to be very irregular and unevenly spaced with several extremely high and low values. Uncertain segments tended to be a mixture of acceptable and unacceptable subsegments with a higher proportion of the former. Time domain features were extracted primarily based on the above observations using peak amplitude and peak position values returned by the heartpy package. Many other extracted features, such as the first and second derivatives, dynamic time warping, Hurst exponent, autocorrelation, and cross-correlation, are popular features useful for time-series data analysis [24,25,26,27]. Frequency domain features were extracted based on the observation that the acceptable segments tend to have similar power frequency spectra, while unacceptable segments tend to have dissimilar power frequency spectra across subsegments. The maxpower, meanfreqdiff, and meanpowerdiff features extracted differences in frequency and power between subsegments based on FFT. In addition, FFT and STFT were applied to extract changes in frequency over time using a Hanning window, where the window size was set to 100 [28].

Future improvements should address the following points. Although *k* is set to 43 in our study, the selection of *k* in the SelectKBest algorithm is best achieved using GridSearch to select the best *k* value that provides the highest precision and recall scores in the held-out training data. Exhaustive and varied feature selection techniques, such as recursive feature elimination and principal component analysis, should be explored [29,30]. Severe data imbalance in the unacceptable and uncertain classes should be addressed using advanced data augmentation techniques specific to time series [31]. Many models were developed during this research that performed very well in terms of precision or recall on any one or two of the three classes. An ensemble approach to classification, such as a voting classifier, may boost accuracy. Valid data augmentation techniques for ECG signals can be applied to increase the sample sizes of unacceptable and uncertain signals. For example, vertical flipping can be a valid augmentation method, as ECG signals frequently flip vertically along the time axis due to sudden changes in polarity. Finally, in the future study, it will be necessary to secure objectivity by increasing the number of physicians for data labeling and to conduct verification on multicenter data.

Even with the very large imbalance in the test dataset, the performance of our approaches was still reasonably good when compared to the performance of other published approaches. Most published work does not deal with more than two classes, and the published approaches used open datasets where imbalance is not a large problem. In papers where imbalance is addressed, the test set is not from an external location.

## 5. Conclusions

There was a very large class imbalance in the quality classification of continuously monitored ECG data. In the classification of ECG quality, where a substantial data imbalance exists, the two-step approaches showed more robust performance than the one-step approach. This algorithm can be used as a preprocessing step in artificial intelligence research using continuously acquired biosignals.

## Figures and Tables

**Figure 1 biomimetics-08-00119-f001:**
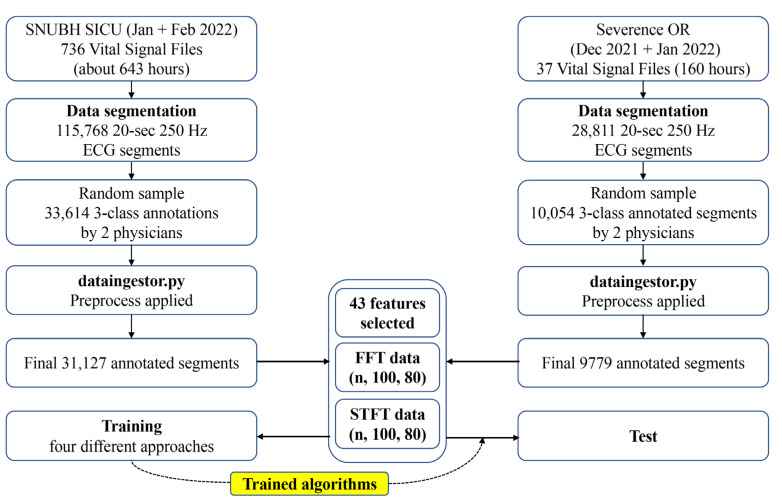
Architecture of the study. SNUBH: Seoul National University Bundang Hospital; ECG: electrocardiography; OR: operating room; FFT: fast Fourier transformation; STFT: short-time Fourier transformation.

**Figure 2 biomimetics-08-00119-f002:**
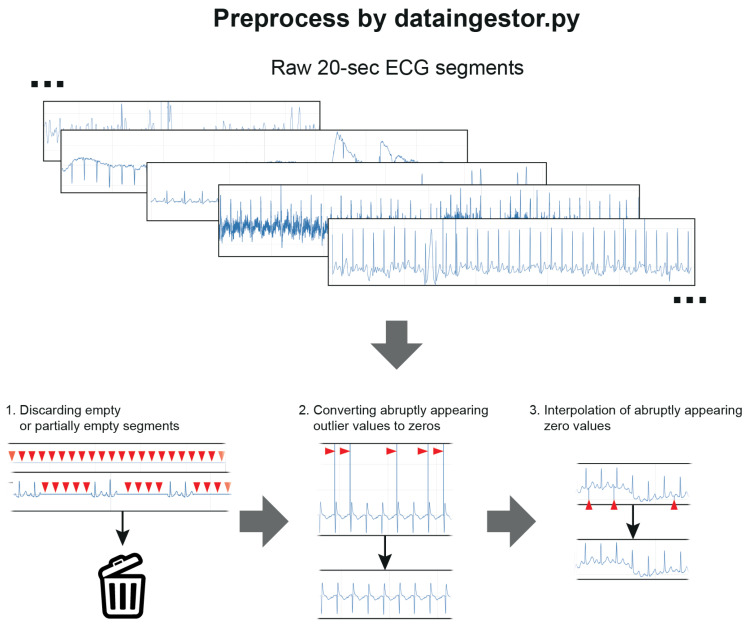
Data cleaning and preprocessing. Red triangles indicate the sites of interest in preprocessing.

**Figure 3 biomimetics-08-00119-f003:**
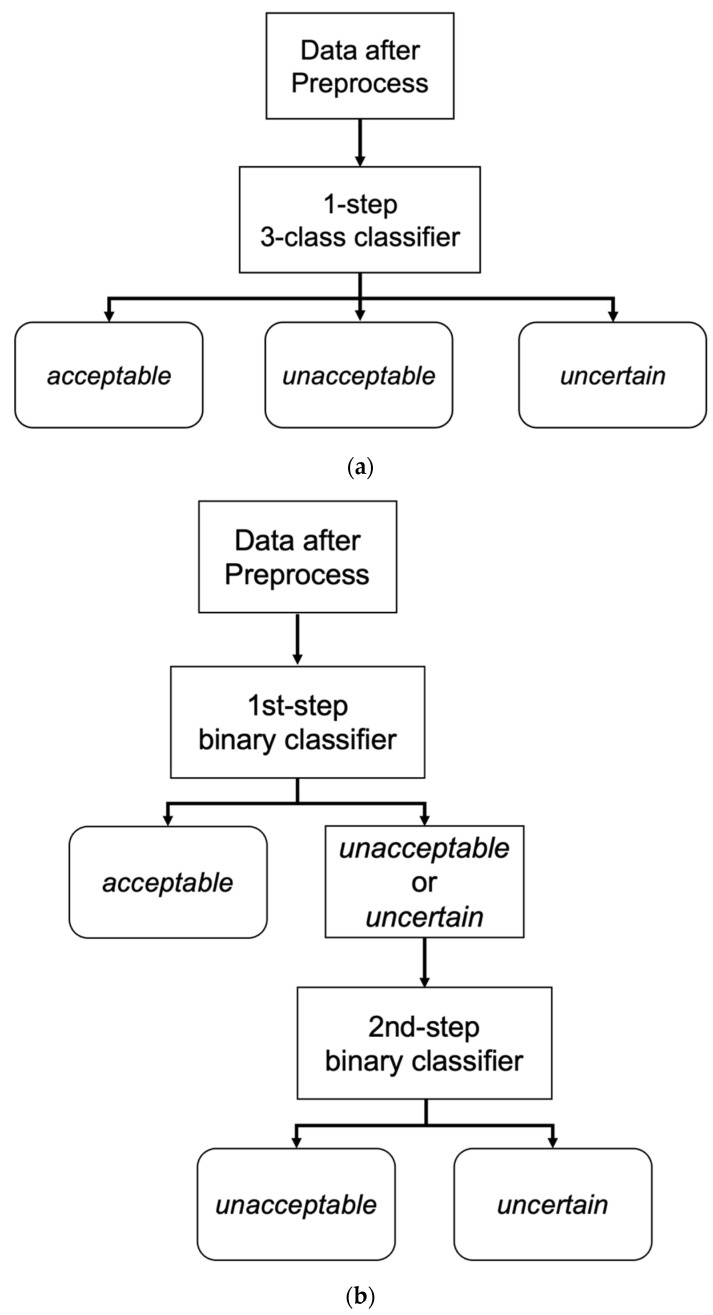
Classification logic of the: (**a**) one-step, 3-class approach; (**b**) two-step binary sequential approach.

**Figure 4 biomimetics-08-00119-f004:**
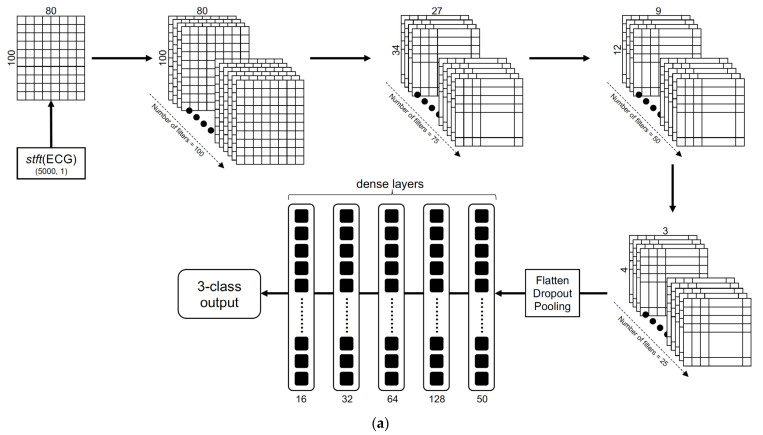
Structure of the 2D CNN classifiers in the one-step and two-step approaches: (**a**) one-step, 3-class 2D CNN classifier; (**b**) a binary 2D CNN classifier in the two-step approach.

**Figure 5 biomimetics-08-00119-f005:**
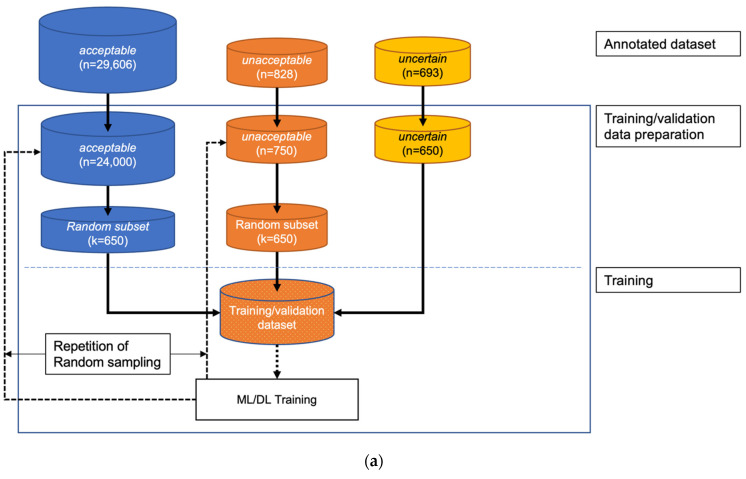
General training scheme of the three classifiers: (**a**) a 3-class classifier in the one-step approach; (**b**) the first-step classifier in the two-step approach; (**c**) the second-step classifier in the two-step approach.

**Figure 6 biomimetics-08-00119-f006:**
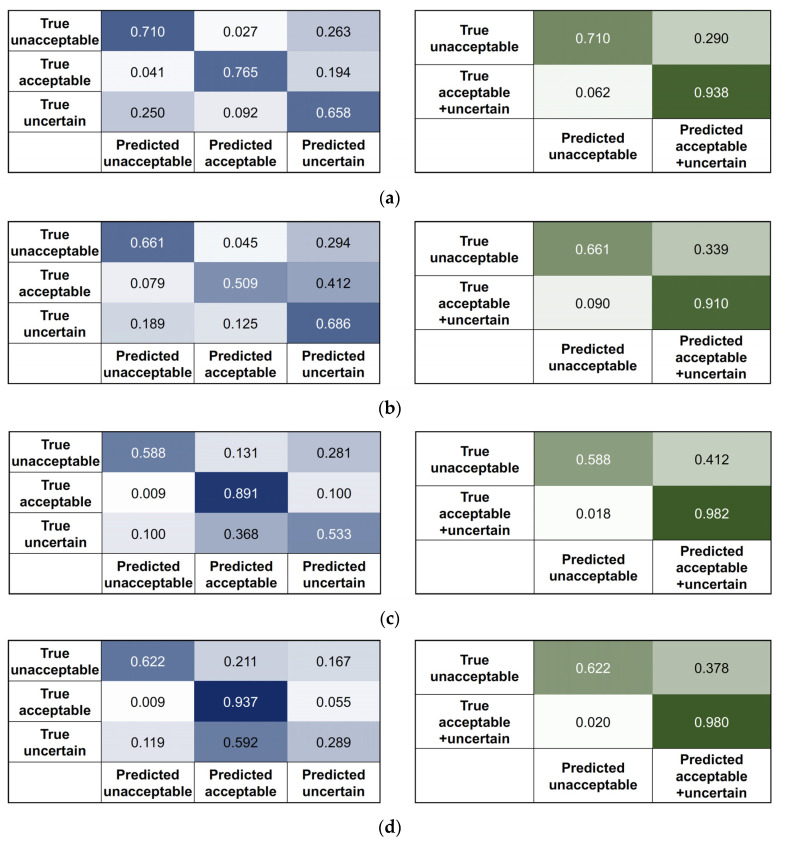
Test results of the one-step approaches on the external data: (**a**) one-step, three-class RF; (**b**) one-step, three-class 2D CNN; (**c**) two-step binary sequential RF approaches; (**d**) two-step binary sequential 2D CNN approaches. The left panels show the 3 × 3 confusion matrices for the original three classes, and the right panels show the 2 × 2 confusion matrices after merging the acceptable and uncertain classes.

**Figure 7 biomimetics-08-00119-f007:**
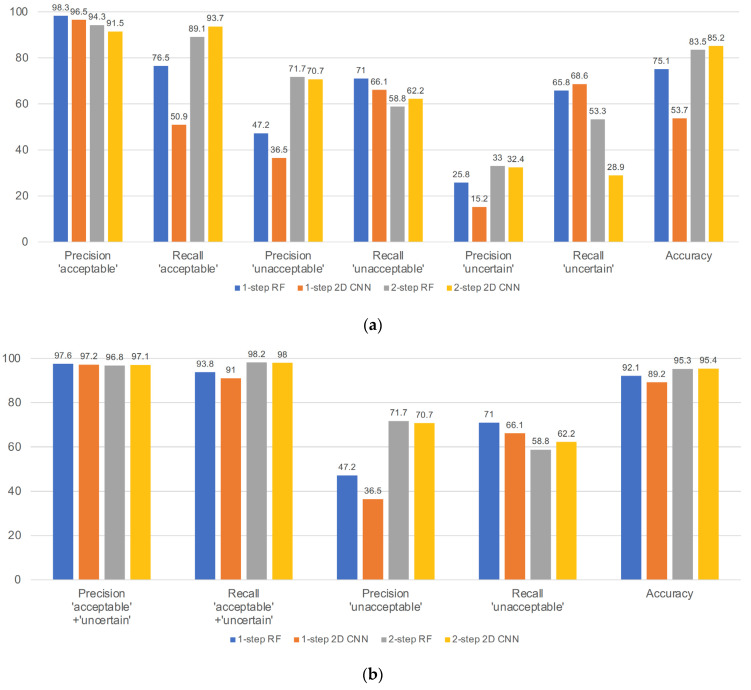
Precision, recall, and accuracy of the 4 approaches: (**a**) performance based on the original three classes; (**b**) performance based on merging the acceptable and uncertain classes. RF: random forest; 2D CNN: two-dimensional convolutional neural network. Unit = %.

**Table 1 biomimetics-08-00119-t001:** List of the 43 features selected using scipy’s KBest algorithm in the RF approaches. Features extracted from the negated signals are indicated with the subscript neg. Features 1–34 are time domain features, while features 35–43 are frequency domain features. The fundamental details and pseudocodes for feature extraction can be found in Appendix A.

No.	Feature Name	Description
1	mean(slopes)	Mean slope
2	stddev(slopes)	Standard deviation of the slope
3	max(zscore(peakamplitude))	Max value of the z score of the peak amplitude
4	min(zscore(peakamplitude_neg_))	Min value of the z score of the peak amplitude obtained from the negated signal
5	max(zscore(peakamplitude_neg_))	Max value of the z score of the peak amplitude obtained from the negated signal
6	max(peakamplitude)	Max value of the peak amplitude
7	max(peakamplitude_neg_)	Max value of the peak amplitude of the negated signal
8	stddev(mean(peakamplitude))	Standard deviation of the mean peak amplitude
9	stddev(max(peakamplitude))	Standard deviation of the max peak amplitude
10	stddev(min(peakminamplitude))	Standard deviation of the min peak amplitude
11	mean(autocorr)	Mean of the auto correlation
12	stddev(autocorr)	Standard deviation of the auto correlation
13	mean(autocorr_neg_)	Mean of the auto correlation of the negated signal
14	stddev(autocorr_neg_)	Standard deviation of the auto correlation of the negated signal
15	mean(crosscorr)	Mean of the cross correlation of 2 nonoverlapping 4 s segments of the signal
16	stddev(crosscorr)	Standard deviation of the cross correlation of 2 nonoverlapping 4 s segments of the signal
17	mean(crosscorr_neg_)	Mean of the cross correlation between 2 nonoverlapping 4 s segments of negated the signal
18	stddev(crosscorr_neg_)	Standard deviation of the cross correlation between 2 nonoverlapping 4 s segments of the negated signal
19	mean(skew)	Mean of the signal skew
20	mean(skew_neg_)	Mean of the negated signal skew
21	stddev(skew)	Standard deviation of the signal skew
22	stddev(skew_neg_)	Standard deviation of the negated signal skew
23	mean(kurtosis)	Standard deviation of the signal kurtosis
24	mean(kurtosis_neg_)	Mean of the negated signal kurtosis
25	stddev(kurtosis)	Standard deviation of the signal kurtosis
26	stddev(kurtosis_neg_)	Standard deviation of the negated signal kurtosis
27	mean(he)	Mean of the Hurst exponent
28	stddev(he)	Standard deviation of the Hurst exponent
29	mean(he_neg_)	Mean of the Hurst exponent of the negated signal
30	stddev(he_neg_)	Standard deviation of the Hurst exponent of the negated signal
31	mean(dtw)	Mean of distances obtained by dynamic time warping
32	stddev(dtw)	Standard deviation of the distances obtained by dynamic time warping
33	mean(dtw_neg_)	Mean of distances obtained by dynamic time warping of the negated signal
34	stddev(dtw_neg_)	Standard deviation of the distances obtained by dynamic time warping of the negated signal
35	mean(maxpower)	Mean of the maximum value of the power obtained after the fast Fourier transform
36	stddev(maxpower)	Standard deviation of the max power obtained after the fast Fourier transform
37	mean(maxpower_neg_)	Mean of the max power obtained after the fast Fourier transform of the negated signal
38	stddev(maxpower_neg_)	Standard deviation of the max power obtained after the fast Fourier transform
39	stddev(meanpowerdiff)	Standard deviation of the maximum difference in the power values
40	stddev(meanfreqdiff)	Standard deviation of the maximum difference in the frequency values
41	mean(abs(stft))	Mean of the absolute value of the short-time Fourier transform
42	std(abs(stft))	Standard deviation of the short-time Fourier transform
43	stddev(se)	Standard deviation of spectral entropy

## Data Availability

Data are available upon reasonable request but need approval from the data committees at SNUBH and SEVH. To request the data from this study, please contact the corresponding author.

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
