# Peer review of "A Two-Step Approach to Overcoming Data Imbalance in the Development of an Electrocardiography Data Quality Assessment Algorithm: A Real-World Data Challenge"

_biomimetics, 2023, doi:10.3390/biomimetics8010119_

Round 1

Reviewer 1 Report

I highly recommend that the authors rewrite the abstract and the conclusion. Instead of listing the performance of the method endless, authors are encouraged to introduce the motivation, method, and general results in the abstract.

Author Response

Comment 1:    “I highly recommend that the authors rewrite the abstract and the conclusion. Instead of listing the performance of the method endless, authors are encouraged to introduce the motivation, method, and general results in the abstract.”

Response 1:     Thank you for your comment. We revised the abstract and conclusion as follows:

Revised abstract:

“…Continuously acquired biosignals from patient monitors contain significant amounts of unusable data. During the development of decision support system based on continuously acquired biosignals, we developed machine and deep learning algorithms to automatically classify the quality of ECG data. A total of 31,127 twenty-second ECG segments of 250 Hz were used as the training/validation dataset. Data quality was categorized into three classes: acceptable, unacceptable, and uncertain. In the training/validation dataset, 29,606 segments (95%) were in the acceptable class. Two 1-step 3-class approaches and two 2-step binary sequential approaches were developed using random forest (RF) and 2-dimensional convolutional neural network (2D CNN) classifiers. Four approaches were tested on 9,779 test samples from another hospital. On the test dataset, the 2-step 2D CNN approach showed the best overall accuracy (0.85), and the 1-step 3-class 2D CNN approach showed the worst overall accuracy (0.54). The most important parameter, precision in the acceptable class, was greater than 0.9 for all approaches, but recall in the acceptable class was better for the 2-step approaches: 1-step (0.77) vs. 2-step RF (0.89) and 1-step (0.51) vs. 2-step 2D CNN (0.94) (P < 0.001 for both comparison). When the acceptable and uncertain classes were merged, all four approaches showed comparable performance, but the 2-step approaches had higher precision in the unacceptable class: 1-step (0.47) vs. 2-step RF (0.72) and 1-step (0.37) vs. 2-step 2D CNN (0.71) (P < 0.001 for both comparison). For ECG quality classification, where substantial data imbalance exists, the 2-step approaches showed more robust performance than the 1-step approach. This result can be used as a preprocessing step in artificial intelligence research using continuously acquired biosignals.” Page 1 lines 20-36

Revised conclusions:

“…There was a very large class imbalance in the quality classification of continuously monitored ECG data. In the classification of ECG quality, where a substantial data im-balance exists, the 2-step approaches showed more robust performance than the 1-step approach. This algorithm can be used as a preprocessing step in artificial intelligence research using continuously acquired biosignals.” Page 16 lines 466-467

Reviewer 2 Report

The paper is well-written and organized. However, I have only a few comments.

1- Please mention the contributions of your work in points clearly at the end of the introduction section and the introduction should be more clear and more elaborative with the proper background.

2- Please add the pseudocode for the proposed algorithm along with the flow chart.

3- Please highlight the limitations of the current work and the motivation for the current work.

I'll be happy to accept the paper after these minor comments.

Author Response

Comment 1:    “Please mention the contributions of your work in points clearly at the end of the introduction section and the introduction should be more clear and more elaborative with the proper background.”

Response 1:     Thank you for your comment. We added a sentence to clarify why we had to study this topic at the end of Introduction, but we did not elaborate too much in the manuscript as it would be beyond the scope of this study.

Added sentence in the Limitation section (end of Discussion section):

“…However, due to the availability of massive biosignal datasets [13-16], researchers have recently used many deep learning techniques in the analysis of various ECG datasets [17-19]. During the process of developing a noninvasive estimator of arterial blood pressure based on ECG and other signals, we needed a signal quality classifier for each signal. This was the most important motivation for this study. In the process of developing a real-time decision support system based on biosignals, we investigated methods for ECG signal quality classification using machine learning and deep learning approaches.” Page 2 lines 73-75

Comment 2:    “Please add the pseudocode for the proposed algorithm along with the flow chart.”

Response 2:     Thank you for your comment. We provided the important real codes in the supplementary material. You can find it in the <Section 6. Source Codes>. Please refer to the supplementary material.

Comment 3:    “Please highlight the limitations of the current work and the motivation for the current work.”

Response 3:     Thank you very much for your comment. The limitations of this work are summarized at the end of Discussion, in the paragraph beginning with “Future improvements should address…”. In that paragraph, we have added a sentence about what we believe to be the most important weak point. As for the motivation for this study, we are currently developing a noninvasive estimator of arterial blood pressure (ABP) waveforms and numbers based on electrocardiography (ECG) and photoplethysmography (PPG) signals. To do this, we need signal quality classifiers for all three signals. This manuscript is a summary of our work on the development of automated ECG quality classifiers. However, our future goal is beyond the scope of this manuscript. We look forward to developing other predictors or estimators that use this ECG quality classifier as part of the front end.

Added sentence in the Introduction:

“…However, due to the availability of massive biosignal datasets [13-16], researchers have recently used many deep learning techniques in the analysis of various ECG da-tasets [17-19]. During the process of developing a noninvasive estimator of arterial blood pressure based on ECG and other signals, we needed a signal quality classifier for each signal. This was the most important motivation for this study. In the process of developing a real-time decision support system based on biosignals, we investigated methods for ECG signal quality classification using machine learning and deep learning approaches.” Page 2 lines 83-85

Added sentence in the Discussion:

“…Valid data augmentation techniques for ECG signals can be applied to increase the sample sizes of unacceptable and uncertain signals. For example, vertical flipping can be a valid augmentation method, as ECG signals frequently flip vertically along the time axis due to sudden changes in polarity. Finally, in the future study, it will be necessary to secure objectivity by increasing the number of physicians for data labeling and to conduct verification on multicenter data.” Page 16 lines 454-456